| LLaVA-Med WA | Med-Flamingo AD | Med-Flamingo WA | MedVInT AD | MedVInT WA | RadFM AD | RadFM WA | LLaVA-v1.6 AD | LLaVA-v1.6 WA | Qwen-VL-Chat AD | Qwen-VL-Chat WA |
|---|---|---|---|---|---|---|---|---|---|---|---|---|
| MIMIC-CXR [17] | **0.10** | **46.14** | 0.68 | 20.58 | _0.13_ | 23.74 | 1.11 | _35.18_ | 0.50 | 32.97 | 0.13 | 23.74 |
| Harvard-FairVLMed [35] | 0.54 | **37.83** | _0.16_ | 21.68 | 0.24 | 27.27 | 0.25 | 35.98 | **0.08** | _37.31_ | 0.25 | 32.93 |
| HAM10000 [45] | 6.81 | 26.52 | _2.22_ | 15.43 | **2.11** | 19.61 | 4.29 | 21.53 | 3.12 | 22.11 | 3.35 | **41.77** |
| OL3I [60] | 3.38 | 28.37 | 3.49 | 32.53 | _0.62_ | **65.64** | 5.21 | 28.20 | 3.84 | 20.36 | **0.33** | 54.12 |

Table 19: Accuracy Equality Difference (%) of LVLMs on demography grouping (the smaller ↓ the better). The best results and second best results are **bold** and underlined, respectively.

| Data Source | MIMIC-CXR [17] Age | MIMIC-CXR [17] Gender | MIMIC-CXR [17] Race | Harvard-FairVLMed [35] Age | Harvard-FairVLMed [35] Gender | Harvard-FairVLMed [35] Race | HAM10000 [45] Age | HAM10000 [45] Gender | OL3I [60] Age | OL3I [60] Gender |
|---|---|---|---|---|---|---|---|---|---|---|
| LLaVA-Med | 8.27 | **0.10** | 7.43 | **5.01** | 0.54 | 2.00 | 14.34 | 6.81 | 45.71 | 3.38 |
| Med-Flamingo | **2.84** | 0.68 | 5.83 | 16.88 | _0.16_ | 3.82 | **7.71** | _2.22_ | **19.75** | 3.49 |
| MedVInT | 5.21 | _0.13_ | **3.08** | 8.98 | 0.24 | _0.96_ | 18.76 | **2.11** | 45.71 | _0.62_ |
| RadFM | 9.52 | 1.11 | 26.73 | _7.86_ | 0.25 | **0.84** | 15.66 | 4.29 | _22.86_ | 5.21 |
| LLaVA-v1.6 | 5.67 | 0.50 | 10.41 | 21.58 | **0.08** | 2.43 | _7.87_ | 3.12 | 43.72 | 3.84 |
| Qwen-VL-Chat | _2.92_ | 0.13 | _4.14_ | 10.54 | 0.25 | 2.13 | 26.85 | 3.35 | 24.00 | **0.33** |

Table 20: Abstention rate (%) of representative LVLMs on overcautiousness evaluation.

| Data Source | LLaVA-Med | Med-Flamingo | MedVInT | RadFM | LLaVA-v1.6 | Qwen-VL-Chat |
|---|---|---|---|---|---|---|
| IU-Xray [5] | 0.61 | 0 | 0 | 0 | 0.03 | 0.02 |
| MIMIC-CXR [18] | 0.54 | 0 | 0 | 0 | 0.05 | 0.02 |
| Harvard-FairVLMed [35] | 0.63 | 0 | 0 | 0.01 | 0.03 | 0.02 |
| HAM10000 [45] | 0.62 | 0 | 0 | 0 | 0.04 | 0.03 |
| OL3I [60] | 0.52 | 0 | 0 | 0.02 | 0.04 | 0.03 |
| PMC-OA [27] | 0.57 | 0 | 0 | 0.01 | 0.04 | 0.05 |
| OmniMedVQA [14] | 0.64 | 0 | 0 | 0.03 | 0.06 | 0.03 |
| Average | 0.59 | 0 | 0 | 0.01 | 0.04 | 0.03 |

Table 21: Performance (%) of six LVLMs based on different "jailbreaking" prompts. Here "Abs": abstention rate, "Acc": accuracy.

| Model | **Concealment** Acc | **Concealment** Abs | **Exaggeration** Acc | **Exaggeration** Abs | **Incorrect Advice** Abs |
|---|---|---|---|---|---|
| LLaVA-Med | 33.73 | 23.62 | 37.49 | 31.74 | 35.15 |
| Med-Flamingo | 21.06 | 0 | 23.88 | 0 | 0 |
| RadFM | 25.82 | 0.19 | 25.04 | 0.44 | 1.32 |
| MedVInT | 33.87 | 0 | 34.33 | 0 | 0 |
| Qwen-VL-Chat | 33.19 | 0.72 | 28.93 | 0.87 | 1.80 |
| LLaVA-v1.6 | 30.12 | 4.14 | 28.64 | 5.52 | 6.42 |

Table 22: Performance (%) of representative LVLMs on toxicity evaluation. Notably, we report the toxicity score (↓) and abstention rate (↑). Here "Tox": toxicity score; "Abs": abstention rate.

| Data Source | LLaVA-Med Tox | LLaVA-Med Abs | Med-Flamingo Tox | Med-Flamingo Abs | MedVInT Tox | MedVInT Abs | RadFM Tox | RadFM Abs | LLaVA-v1.6 Tox | LLaVA-v1.6 Abs | Qwen-VL-Chat Tox | Qwen-VL-Chat

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

These potential social risks warrant attention to encourage the emergence of reliable Med-LVLMs in the future.

## I   Data Sheet

We follow the documentation frameworks provided by Wang et al. [49].

### I.1   Motivation

**For what purpose was the dataset created?**

- Our benchmark aims to comprehensively evaluate the trustworthiness of Med-LVLMs. This study provides valuable references and foundations for the reliable development of Med-LVLMs and the deployment of future models in real clinical settings. We primarily assess trustworthiness from the following five perspectives: *trustfulness, fairness, safety, privacy, and robustness*.

**Who created the dataset (e.g., which team, research group) and on behalf of which entity (e.g., company, institution, organization)?**

- Our dataset is jointly developed by a collaborative effort from the following research groups:
    - The University of North Carolina at Chapel Hill (UNC-Chapel Hill)
    - Stanford University
    - University of Illinois at Urbana-Champaign (UIUC)
    - Brown University
    - University of Washington
    - Microsoft Research
    - The University of Texas at Arlington (UT Arlington)
    - Monash University

## I.2  Composition/collection process/preprocessing/cleaning/labeling and uses:

- The answers are described in our paper as well as website https://github.com/richard-peng-xia/CARES.

## I.3  Distribution

**Will the dataset be distributed to third parties outside of the entity (e.g., company, institution, organization) on behalf of which the dataset was created?**

- No. Our dataset will be managed and maintained by our research group.

**How will the dataset will be distributed (e.g., tarball on website, API, GitHub)?**

- The evaluation dataset is released to the public and hosted on GitHub.

**When will the dataset be distributed?**

- It has been released now.

**Will the dataset be distributed under a copyright or other intellectual property (IP) license, and/or under applicable terms of use (ToU)?**

- Our dataset will be distributed under the CC BY-SA 4.0 license.

## I.4  Maintenance

**How can the owner/curator/manager of the dataset be contacted (e.g., email address)?**

- Please contact Peng Xia (richard.peng.xia@gmail.com) and Prof. Huaxiu Yao (huaxiu@cs.unc.edu), who are responsible for maintenance.

**Will the dataset be updated (e.g., to correct labeling errors, add new instances, delete instances)?**

- Yes. We will make announcements on GitHub if there is any update.

**Is there an erratum?**

- No. We will make it if there is any erratum.

**If the dataset relates to people, are there applicable limits on the retention of the data associated with the instances (e.g., were individuals in question told that their data would be retained for a fixed period of time and then deleted)?**

- N/A.

**If others want to extend/augment/build on/contribute to the dataset, is there a mechanism for them to do so?**

- For dataset contributions and evaluation modifications, the most efficient way to reach us is via GitHub pull requests.

- For more questions, please contact Peng Xia (`richard.peng.xia@gmail.com`) and Prof. Huaxiu Yao (`huaxiu@cs.unc.edu`), who will be responsible for maintenance.