# OpenReview forum: "CARES: A Comprehensive Benchmark of Trustworthiness in Medical Vision Language Models"
_NeurIPS.cc/2024/Datasets_and_Benchmarks_Track — NeurIPS 2024 Track Datasets and Benchmarks Poster_

### Official Review · Reviewer_xYtq · 2024-06-21
**Interesting topic but evaluations are flawed**

**Rating:** 6
**Confidence:** 4
**Correctness:** I think many evaluation setups are fl…
**Clarity:** Yes.

**Review:**

## Pros:
- This paper studies a crucial topic, trustworthiness, in the deployment of VLMs in medical imaging. The paper is well motivated.
- CARES assesses trustworthiness across five dimensions (trustfulness, fairness, safety, privacy, and robustness), and include a large number of question-answer pairs and modalities. It can be a useful resource for the community.
- Different models are evaluated on this benchmark and the authors provide interesting insights on the results.
- The paper is clearly written and well organized.

## Cons:

### Concerns on general evaluation

1. For open-ended questions, in section 2, the authors mentioned that "We request GPT-4 to rate the helpfulness, relevance, accuracy, and level of detail of the ground-truth answers and model responses and provide an overall score ranging from 1 to 10". This is perhaps acceptable for general performance evaluation (although GPT4 can still make mistakes), but since this paper focuses on trustworthiness, the evaluation results will be biased towards GPT4's ``opinion''. At the same time, GPT4 is not evaluated on the proposed benchmark, so we don't know how much GPT4 is biased.
2. For close-ended questions, many binary classification tasks are transformed into yes/no questions, and accuracy is used as the main metrics. For medical imaging, this can be a severe problem as the label distribution is severely imbalanced, including the datasets used in this paper. For example, if 90% of labels are benign, the model can easily get a 90% Acc by predicting everything to 0. That's why AUROC is more widely used.
3. The total number of evaluated models are limited. Some state-of-the-art vision-language models, even not medical fine-tuned, should still be strong in this setting, such as GPT4, VILA, InternLM-XComposer2, and more.

### Concerns on specific dimensions

1. Uncertainty on Trustfulness: the authors prompt the model with "are you sure you accurately answered the question" to evaluate uncertainty. This approach is biased itself because the answer to this question do not necessarily represent the confidence of the original prediction and also highly depend on how they are aligned, which is independent of the real questions, c.f. [1]. Also, a binary yes/no cannot quantify the level of confidence. I think probability-based calibration is still preferred.
2. For table 1, related to general concern 2, some models have very high performance on one dataset while low performance on other datasets, while some models show a different trend. E.g., RadFM v.s. Qwen-VL on IU-Xray and OL3I. I strongly suspect this is due to the strong language prior and imbalanced label distribution. For example, some model may tend to answer "yes" rather than "no". Therefore, for labels that are mostly yes, some model will get very high performance and vice versa. The authors can check the predictions and labels to see if it's true.
3. Again, for fairness evaluation, it's shown in previous study that AUROC is much preferred than Acc [2]. Also, it will be helpful to consider Max-Min Fairness as well.
4. For safety, I think it's debatable that whether the jailbreaking prompts are really meaningful in practice, as we can from the input (e.g., "give an exaggerated answer") that the generated predictions are not useful. This is different from the general jailbreaking like ``how to make a bomb?'' because malicious users can really use the outputs for harmful knowledge.
5. For privacy, I'm wondering how is it possible for the models to predict the marital status and social security number from images? If not, I think this section is more like a jailbreaking test that belongs to ``safety'' section because the models do not know the PHI thus no privacy leakage.
6. For robustness, do we want the models to reject to answer questions of the noisy images (abstention rate in Table 5)? How do the authors judge whether it is "too noisy for making accurate"? Since the accuracy degradation is not too significant (usually <10%), we may want the model to answer the questions, which can be useful in scenarios where the images are not well captured.




[1] Language Models Don’t Always Say What They Think: Unfaithful Explanations in Chain-of-Thought Prompting. NeurIPS 2023. \
[2] MEDFAIR: Benchmarking Fairness for Medical Imaging. ICLR 2023.

**Strengths:**

See pros above.

**Additional Feedback:**

See above.

**Documentation:**

The provided Github is empty at the time of writing this review and the dataset is also not released.

**Limitations:**

Limitations not discussed.

**Opportunities For Improvement:**

See cons above.

**Relation To Prior Work:**

Yes.

**Summary And Contributions:**

The paper introduces CARES, a comprehensive benchmark designed to evaluate the trustworthiness of Medical Large Vision Language Models across five dimensions: trustfulness, fairness, safety, privacy, and robustness. The benchmark comprises 41K question-answer pairs from 16 medical image modalities and 27 anatomical regions, revealing significant trustworthiness concerns in current Med-LVLMs, including factual inaccuracies, demographic biases, privacy breaches, and susceptibility to attacks.

---

> ### Author Rebuttal · Authors · 2024-08-17
>
> Thank you for your valuable feedback and suggestions.
>
> >**Q1**: Potential bias of GPT4.
>
> **A1**: Regarding the evaluation of open-ended questions, since open-ended VQA lacks precise evaluation metrics and common language quality evaluation metrics (e.g., BLEU, ROUGE) do not fully capture medical factuality, therefore, we adopt an automated metric [1] based on GPT-4 to score the ground-truth answers and model responses. To make the evaluation more reproducible, we follow the POPE [2] by inputting medical reports into GPT-4 to convert them into closed-ended VQA, with answers limited to "yes" or "no." In summary, as shown in Table R1 in the attached PDF, all evaluated models still exhibit significant factuality hallucination.
>
> ***
>
> >**Q2**: For close-ended questions, binary classification tasks are … AUROC is more widely used.
>
> **A2**: We have computed the label distributions for the only binary classification dataset (i.e., OL3I), as shown in Table R2 in the attached PDF. In light of your suggestion, in addition to accuracy, as shown in Table R3 in the attached PDF, we also use F1 score and AUROC to evaluate model performance, and we report the proportion of "yes" responses to check for the potential influence of language priors. The results across multiple metrics indicate that the existing models suffer from significant factuality hallucination, and their accuracy does not rely on label distribution.
>
> ***
>
> >**Q3**: The total number of evaluated models is limited.
>
> **A3**: We compare several of the latest state-of-the-art LVLMs and Med-LVLMs, including VILA, InternLM-XComposer2, InternVL2 and LLaVA-Med-v1.5, with the results presented in Table R4 in the attached PDF. Although advanced Med-LVLMs outperform other Med-LVLMs and state-of-the-art LVLMs, they still exhibit trustworthiness issues.
>
> ***
>
> >**Q4**: Uncertainty on Truthfulness.
>
> **A4**: In light of your suggestion, we adopt a probability-based approach for uncertainty estimation [3], using the model's produced conditional probabilities to determine its confidence levels. We then use ECE, AUROC, and MSE [4,5] as evaluation metrics. As shown in Table R5 in the attached PDF, the results align with the conclusions from the prompt-based testing approach, indicating that the current Med-LVLMs exhibit poorly calibrated uncertainty.
>
> ***
>
> >**Q5**: Language prior and imbalanced label distribution.
>
> **A5**: We have computed the label distributions for the IU-Xray and OL3I closed-ended VQA datasets, as shown in Table R6 in the attached PDF. In response to your concerns, we have also provided the evaluation results for RadFM and Qwen-VL-Chat on these two datasets, as shown in Table R7 in the attached PDF. The results indicate that the models' answers are relatively reasonable and do not excessively rely on language priors or label distributions. The performance differences between models on different datasets is likely due to the reflect the general exposure to data modalities or linguistic characteristics during the training stages.
>
> ***
>
> >**Q6**: Adopt the AUROC and Max-Min Fairness.
>
> **A6**: For the closed-ended VQA dataset derived from classification datasets, we will add AUROC results to mitigate the impact of class imbalance on the metrics. We have already implemented Max-Min Fairness, and in Tables 18 and 19 of the Appendix (Page 13), we present the results for Demographic Accuracy Difference and Worst Accuracy, aiming to provide a comprehensive assessment of the model's fairness.
>
> ***
>
> >**Q7**: Jailbreaking prompts for safety.
>
> **A7**: Conducting prompt-based attacks on Med-LVLM is essential for evaluating the potential risks these models may face. Here, we leverage a simple jailbreaking test to evaluate the model's safety capabilities. We acknowledge that these prompt-based attacks are relatively straightforward and easily identifiable. However, even with these simple tests, the abstention rate of most models was less than 10%, indicating significant safety vulnerabilities. In light of your suggestion, we will implement more complex jailbreaking tests in scenarios. For example, if data is available, we can use prompt-based attacks to induce the model to generate incorrect recommendations, such as repeating ineffective tests or omitting critical medications.
>
> ***
>
> >**Q8**: Setup in privacy evaluation.
>
> **A8**: Regarding privacy evaluation, we have referenced some trustworthiness evaluation work [6] in the field of LLMs. Privacy evaluation first examines whether the model refuses to answer questions involving private information. If the model refuses to respond, there is no privacy breach. However, the abstention rate for most models is less than 10%. We then assess whether the model's responses leak private information by evaluating the accuracy of its answers regarding marital status. As shown in Table 4 in the paper, the accuracy of Med-Flamingo and RadFM in answering marital status questions is around 50%, which clearly exceeds the results of random guessing (Acc 36.42%). This indicates that they indeed reveal users' marital status in their responses, raising privacy concerns.
>
> ***
>
> >**Q9**: Setup in robustness evaluation.
>
> **A9**: From the results in Table 5 in the paper, we can observe that when confronted with noisy images, the model's accuracy generally decreases by about 10%. Additionally, in light of your suggestions in Q2 and Q5, we report the AUC decline in Table R8, which shows a similar trend to the decline in accuracy. We will revise our claim to focus more on the extent of performance degradation rather than on the abstention rate.
>
> ***
>
> **Reference**
>
> [1] LLaVA-Med: Training a … One Day.
>
> [2] Evaluating Object Hallucination in Large Vision-Language Models.
>
> [3] Just Ask for Calibration: … with Human Feedback.
>
> [4] Can LLMs Express Their Uncertainty? … Elicitation in LLMs.
>
> [5] Semantic Uncertainty: … in Natural Language Generation.
>
> [6] DecodingTrust: A Comprehensive Assessment of Trustworthiness in GPT Models.

---

> > ### Author Response · Authors · 2024-08-22
> > **We would like to hear back from reviewer xYtq**
> >
> > Dear Reviewer xYtq,
> >
> > We wish to once again express our heartfelt appreciation for the time you took to review our paper.
> >
> > Given the short duration of the author-reviewer discussion period, we would greatly appreciate your feedback on whether your main concerns have been adequately addressed. We are always happy to provide further explanations or empirical results whenever necessary.
> >
> > Thank you very much!

---

> > > ### Comment · Reviewer_xYtq · 2024-08-23
> > >
> > > Thank you very much for your efforts. Most of my concerns are addressed and I'm raising the score to 6.
> > >
> > > For jailbreaking experiment, as the authors said, this would be a much better alternative compared to the original one. I'd recommend incorporating this into the final version.
> > > > We will implement more complex jailbreaking tests in scenarios. For example, if data is available, we can use prompt-based attacks to induce the model to generate incorrect recommendations, such as repeating ineffective tests or omitting critical medications.

---

### Official Review · Reviewer_MGa1 · 2024-07-03
**Review Comments**

**Rating:** 7
**Confidence:** 3
**Correctness:** I think the benchmark is constructed …
**Clarity:** It is well-written.

**Review:**

Quality: The paper demonstrates high quality in its comprehensive approach to evaluating Med-LVLMs. The authors have developed a robust benchmark that covers multiple dimensions of trustworthiness, using a large and diverse dataset. The experimental design appears sound, with clear metrics and evaluation procedures.

Clarity: The paper is generally well-structured and clearly written. The authors effectively explain their methodology, the dimensions of trustworthiness they evaluate, and their findings. The use of figures and tables helps illustrate key points and results.

Originality: CARES represents a novel contribution to the field. While there have been evaluations of Med-LVLMs before, this paper's comprehensive approach covering five dimensions of trustworthiness is innovative. The introduction of new metrics, such as the overconfidence ratio, also adds to its originality.

Significance: This work is highly significant for the field of medical AI. As Med-LVLMs become increasingly prevalent in healthcare applications, a thorough understanding of their trustworthiness is crucial. CARES provides a standardized benchmark that can guide future development and evaluation of these models, potentially improving their reliability and safety in clinical settings.

Pros:

1. The paper introduces a holistic approach to evaluating Med-LVLMs by assessing five critical dimensions of trustworthiness: trustfulness, fairness, safety, privacy, and robustness. This multifaceted evaluation provides a more complete picture of model performance and reliability compared to previous studies that often focused on single dimensions.

2. CARES includes a large and diverse dataset comprising about 41,000 question-answer pairs, covering 16 medical image modalities and 27 anatomical regions. This extensive coverage allows for a more thorough and representative evaluation of Med-LVLMs across various medical specialties and image types.

3. By evaluating multiple Med-LVLMs and general LVLMs using CARES, the paper provides valuable insights into the current state of these models' trustworthiness. This benchmarking effort not only highlights areas for improvement but also sets a standard for future development and evaluation of Med-LVLMs, potentially driving progress towards more reliable and safe AI systems in healthcare.

Cons:

1. It would be better to include a comparison with human expert performance on the same tasks. Such a comparison could provide context for how the AI models' performance relates to current clinical standards and help identify areas where AI might already be meeting or exceeding human-level performance.

Minor point:
2. The paper evaluates only four open-source Med-LVLMs and two general LVLMs. While this provides valuable insights, it may not fully represent the entire landscape of medical AI models, particularly proprietary or more recently developed ones.

3. While the paper effectively identifies issues in current Med-LVLMs, it provides limited discussion on potential strategies to mitigate these problems. More insights into possible solutions or directions for improvement could enhance the paper's impact.

4. The paper presents a snapshot of current model performance but doesn't include an analysis of how model trustworthiness has evolved over time or with different versions. Such an analysis could provide insights into the trajectory of improvements in Med-LVLMs and help predict future trends.

**Strengths:**

Significance of Contribution: Provides a standardized framework for assessing model reliability and safety in healthcare applications

Relevance to Broader Research Community: Contributes to the broader discourse on AI safety, reliability, and ethical implementation

Quality of Research: Utilizes a large, diverse dataset covering 16 medical image modalities and 27 anatomical regions; Employs a multidimensional evaluation approach (trustfulness, fairness, safety, privacy, robustness)

Ethical and Social Implications: Addresses critical ethical concerns in AI-assisted healthcare; Fairness evaluation across demographic groups; Assessment of privacy protection capabilities; Examination of model safety and robustness

**Additional Feedback:**

Please refer to the Cons in the review section.

**Documentation:**

It provides the code intended to reproduce the experiments in this work, but I have not tested it due to the limited time frame.

**Ethics:**

I don't have any ethical concerns with this work.

**Limitations:**

Please refer to the Cons in the review section.

**Opportunities For Improvement:**

Please refer to the Cons in the review section.

**Relation To Prior Work:**

This work clearly discusses how it differs from previous contributions.

**Summary And Contributions:**

This paper introduces CARES, a comprehensive benchmark for evaluating the trustworthiness of Medical Large Vision Language Models (Med-LVLMs) across five key dimensions: trustfulness, fairness, safety, privacy, and robustness. The benchmark comprises about 41,000 question-answer pairs covering 16 medical image modalities and 27 anatomical regions. The authors evaluate several Med-LVLMs and general LVLMs using CARES, revealing significant concerns regarding trustworthiness, including factual inaccuracies, fairness issues across demographic groups, vulnerability to attacks, lack of privacy awareness, and poor out-of-distribution robustness. The study aims to drive the development of more reliable Med-LVLMs and standardize their evaluation.

---

> ### Author Rebuttal · Authors · 2024-08-17
>
> Thank you for your constructive comments. We respond to your questions below and would appreciate it if you could let us know if our response addresses your concerns.
>
> >**Q1**: It would be better to include a comparison with human expert performance on the same tasks. Such a comparison could provide context for how the AI models' performance relates to current clinical standards and help identify areas where AI might already be meeting or exceeding human-level performance.
>
> **A1**: We have already engaged radiology experts to begin the manual evaluation of radiology-related content. Additionally, we plan to involve more multidisciplinary medical professionals (pathology, ophthalmology, etc.) to enhance human expert performance in the CARES benchmark.
>
> ***
>
> >**Q2**: The paper evaluates only four open-source Med-LVLMs and two general LVLMs. While this provides valuable insights, it may not fully represent the entire landscape of medical AI models, particularly proprietary or more recently developed ones.
>
> **A2**:  As LVLMs continue to develop rapidly, the number of models we evaluate will further increase. We have compared several of the latest state-of-the-art LVLMs and Med-LVLMs, including VILA [1], InternLM-XComposer2 [2], InternVL2 [3] and LLaVA-Med-v1.5, with the results presented in Table R1. Although advanced Med-LVLMs outperform other Med-LVLMs and state-of-the-art LVLMs, they still exhibit trustworthiness issues.
>
> **Table R1**: Model performance on factuality evaluation.
> | Dataset             | LLaVA-Med-v1.0 | Med-Flamingo | MedVInT | RadFM | LLaVA-v1.6 | Qwen-VL-Chat | LLaVA-Med-v1.5 | VILA  | InternLM-XComposer2 | InternVL2 |
> |---------------------|----------------|--------------|---------|-------|------------|---------------|----------------|-------|------------------------|------------|
> | MIMIC-CXR        | 71.13          | 61.27        | 66.06   | 69.30 | 63.70      | 60.43         | **75.79**      | 62.50 | 60.17                  | 59.47      |
> | Harvard-FairVLMed| 61.37          | 42.06        | 35.92   | 52.47 | 48.52      | 38.06         | **63.03**      | 54.08 | 47.61                  | 44.38      |
>
> ***
>
> >**Q3**: While the paper effectively identifies issues in current Med-LVLMs, it provides limited discussion on potential strategies to mitigate these problems. More insights into possible solutions or directions for improvement could enhance the paper's impact.
>
> **A3**: We will include a more extensive discussion on methods to mitigate the trustworthiness issues of Med-LVLMs in the discussion section. For example, to improve factuality, we can retrieve the medical descriptions most relevant to the input image as auxiliary prompts, or we can fine-tune the model using high-quality data for downstream tasks. To improve safety, we can fine-tune the model using the data specifically designed for safety to align the behavior of the model. To improve fairness, we need to check the demographic information distribution of the training data and then fine-tune the model in the consideration of data distribution to avoid the group bias.
>
> ***
>
> >**Q4**: The paper presents a snapshot of current model performance but doesn't include an analysis of how model trustworthiness has evolved over time or with different versions. Such an analysis could provide insights into the trajectory of improvements in Med-LVLMs and help predict future trends.
>
> **A4**: We conduct trustworthiness evaluation for different versions of the model to analyze how the model's trustworthiness changes over time or versions. Among the existing Med-LVLMs, LLaVA-Med is the most representative, and the recent release of LLaVA-Med-v1.5 has further advanced the performance of LLaVA-Med-v1.0. To understand the impact of model updates, we compare the performance of LLaVA-Med-v1.5 and LLaVA-Med-v1.0 in factuality and fairness, as shown in Table R2 and Table R3, respectively. The results indicate that the improved LLaVA-Med has shown advancements on the accuracy of VQA tasks, but a decline in fairness evaluation. One potential reason is that the model's improvement strategy does not consider fairness, focusing primarily on factual accuracy. In fact, this is also the focus of most current model improvement strategies. Trustworthiness evaluations can help identify such issues and guide the future direction of model development.
>
> **Table R2**: Longitudinal comparison on factuality evaluation with LLaVA-Med.
> | Dataset             | LLaVA-Med-v1.0 | LLaVA-Med-v1.5 | Med-Flamingo | MedVInT | RadFM | LLaVA-v1.6 | Qwen-VL-Chat |
> |---------------------|----------------|----------------|--------------|---------|-------|------------|---------------|
> | MIMIC-CXR       | 71.13          | **75.79**          | 61.27        | 66.06   | 69.30 | 63.70      | 60.43         |
> | Harvard-FairVLMed| 61.37          | **63.03**          | 42.06        | 35.92   | 52.47 | 48.52      | 38.06         |
>
> **Table R3**: Longitudinal comparison on fairness evaluation (gender) with LLaVA-Med. We report the performance using Accuracy Equality Difference (%) $\downarrow$.
> | Dataset             | LLaVA-Med-v1.0 | LLaVA-Med-v1.5 | Med-Flamingo | MedVInT | RadFM | LLaVA-v1.6 | Qwen-VL-Chat |
> |---------------------|----------------|----------------|--------------|---------|-------|------------|---------------|
> | MIMIC-CXR        | **0.10**       | 0.12           | 0.68         | 0.13    | 1.11  | 0.50       | 0.13          |
> | Harvard-FairVLMed| 0.54           | 0.55           | 0.16         | 0.24    | 0.25  | **0.08**   | 0.25          |
>
> ***
> **Reference**:
>
> [1] VILA: On Pre-training for Visual Language Models.
>
> [2] Internlm-xcomposer-2.5: A versatile large vision language model supporting long-contextual input and output.
>
> [3] How far are we to gpt-4v? closing the gap to commercial multimodal models with open-source suites.

---

### Official Review · Reviewer_8Xi4 · 2024-07-12
**CARES: A Comprehensive Benchmark of Trustworthiness in Medical Vision Language Models**

**Rating:** 6
**Confidence:** 4
**Correctness:** Yes
**Clarity:** Yes

**Review:**

Please see below.

**Strengths:**

The paper is easy-to-follow. Although Med-VLMs have shown excellent performance in medical diagnosis and other applications, their trustworthiness has not been fully validated, which poses significant risks for future model deployment. Existing evaluations often focus on only specific dimensions of Med-VLMs' trustworthiness, lacking a systematic and comprehensive assessment. Therefore, this paper proposes the CARES benchmark to comprehensively evaluate the trustworthiness of Med-VLMs from multiple dimensions, aiming to promote the design of more reliable Med-VLMs in the future.

**Additional Feedback:**

N/A

**Documentation:**

Yes

**Ethics:**

Yes

**Limitations:**

Please see above.

**Opportunities For Improvement:**

1. Data quality depends on GPT-4. The construction of open-ended Q&A pairs and manual evaluation in CARES rely on GPT-4. However, GPT-4 itself is not perfect, which may affect the quality of data and evaluation. In the future, more human review could be considered to improve annotation reliability.
2. Limited evaluation dimensions. Although CARES already evaluates the trustworthiness of Med-LVLMs from multiple dimensions, there is still room for improvement. For example, more types of privacy data could be included in privacy evaluation, and more attack scenarios could be designed in safety evaluation. Additionally, ethical and causal reasoning abilities are also worthwhile trustworthiness dimensions to consider.
3. Lack of analysis of internal model mechanisms. CARES mainly focuses on output-level evaluation of models, lacking in-depth analysis of internal knowledge representation and reasoning mechanisms. Combining internal interpretability analysis might better understand the reasons behind model behaviors and provide more insights for improvement.
4. Lack of longitudinal comparison. CARES currently mainly compares the performance of different Med-LVLMs horizontally, lacking longitudinal analysis of trustworthiness changes in the same model during updates and iterations. Longitudinal analysis helps understand the impact of model scale, data, algorithms, and other factors on trustworthiness improvement.
5. Update and iteration of the evaluation benchmark. With the rapid development of Med-LVLMs, CARES needs to update evaluation tasks, data, and metrics to reflect the latest progress and needs in the field. Establishing an active community to continuously maintain and optimize the evaluation benchmark will be important work.
6. Limited clinical application guidance from evaluation results. Currently, CARES evaluation results mainly serve researchers and model developers, with limited direct guidance for clinical applications. In the future, some key indicators that help guide clinical deployment could be refined, and usage guidelines for doctors could be provided.
7. Ignores the human-machine collaboration perspective. CARES mainly evaluates the performance of Med-LVLMs as standalone medical assistance tools, ignoring the human-machine collaboration perspective. In practice, Med-LVLMs often serve as assistants to doctors, and it's necessary to consider how to evaluate the model's utility in assisting medical decision-making and improving doctors' work efficiency.
8. Lack of consideration for long-term impacts. The current CARES focuses on the current performance of Med-LVLMs, lacking consideration of potential long-term impacts, such as effects on doctor-patient relationships and medical resource allocation. Incorporating perspectives from more stakeholders and examining the trustworthiness of Med-LVLMs from a longer-term and more comprehensive perspective would be valuable.

**Relation To Prior Work:**

Yes

**Summary And Contributions:**

CARES is a comprehensive benchmark for evaluating the trustworthiness of Medical Vision-Language Models (Med-VLMs) from five dimensions: trustfulness, fairness, safety, privacy, and robustness. Using a large-scale medical multimodal dataset (covering 16 imaging modalities and 27 anatomical regions), it systematically evaluates the performance of representative Med-VLMs. Results show that existing Med-VLMs have deficiencies in all trustworthiness dimensions: poor factuality and uncertainty judgment abilities, significant demographic biases, lack of defense against "jailbreak" attacks and toxic outputs, weak privacy protection awareness, and lack of robustness to out-of-distribution samples. CARES reveals many limitations of Med-VLMs in terms of trustworthiness, providing important insights and references for the development of more reliable Med-VLMs in the future. Meanwhile, the construction of CARES itself has some shortcomings, such as dependence on GPT-4 and limitations in evaluation dimensions, which need continuous iteration and optimization to keep up with the latest developments in Med-VLM technology.

---

> ### Author Rebuttal · Authors · 2024-08-17
>
> Thank you for reviewing our paper and for your valuable feedback. Below, we address your concerns point by point. We would appreciate it if you could let us know whether your concerns are addressed by our response.
>
> >**Q1**: Data quality depends on GPT-4.
>
> **A1**: The current construction of open-ended VQA leverages GPT-4's strong text reasoning and generation capabilities. Nevertheless, we acknowledge that GPT-4 may make errors in a few scenarios, so, in light of your suggestion, we plan to introduce human review (including a group of medical specialists from various medical departments) in the future to enhance the reliability of the generated data.
>
> ***
>
> >**Q2**: Limited evaluation dimensions.
>
> **A2**: The evaluation methods for the five dimensions of trustworthiness in CARES follow common practice in LLMs [1].  Meanwhile, we agree more dimensions can be considered to enrich the evaluation in CARES and we plan to add more in the next version. For instance, as you suggested, ethics and causal reasoning will be the focus of our next evaluation phase.
>
> ***
>
> >**Q3**: Lack of analysis of internal model mechanisms.
>
> **A3**: CARES primarily focuses on evaluating the outputs generated by the models, as the outputs most directly reflect the model's performance and are easier to quantify and compare. In the future, we will conduct an in-depth analysis of the model's internal knowledge representation and reasoning mechanisms, thereby providing a more detailed assessment of the model's performance from an interpretability perspective.
>
> ***
>
> >**Q4**: Lack of longitudinal comparison.
>
> **A4**: Among the existing Med-LVLMs, LLaVA-Med is the most representative, and the recent release of LLaVA-Med-v1.5 has further advanced the performance of LLaVA-Med-v1.0. To understand the impact of model updates, we compare the performance of LLaVA-Med-v1.5 and LLaVA-Med-v1.0 in factuality and fairness, as shown in Table R1 and Table R2, respectively. The results indicate that the improved LLaVA-Med has shown advancements on the accuracy of VQA tasks, but a decline in fairness evaluation. One potential reason is that the model's improvement strategy does not consider fairness, focusing primarily on factual accuracy. In fact, this is also the focus of most current model improvement strategies. Trustworthiness evaluations can help identify such issues and guide the future direction of model development, thereby promoting the reliable growth of the Med-LVLM community.
>
> **Table R1**: Longitudinal comparison on factuality evaluation with LLaVA-Med.
> | Dataset             | LLaVA-Med-v1.0 | LLaVA-Med-v1.5 | Med-Flamingo | MedVInT | RadFM | LLaVA-v1.6 | Qwen-VL-Chat |
> |---------------------|----------------|----------------|--------------|---------|-------|------------|---------------|
> | MIMIC-CXR        | 71.13          | **75.79**          | 61.27        | 66.06   | 69.30 | 63.70      | 60.43         |
> | Harvard-FairVLMed | 61.37          | **63.03**          | 42.06        | 35.92   | 52.47 | 48.52      | 38.06         |
>
>
> **Table R2**: Longitudinal comparison on fairness evaluation (gender) with LLaVA-Med. We report the performance using Accuracy Equality Difference (%) $\downarrow$.
> | Dataset             | LLaVA-Med-v1.0 | LLaVA-Med-v1.5 | Med-Flamingo | MedVInT | RadFM | LLaVA-v1.6 | Qwen-VL-Chat |
> |---------------------|----------------|----------------|--------------|---------|-------|------------|---------------|
> | MIMIC-CXR        | **0.10**       | 0.12           | 0.68         | 0.13    | 1.11  | 0.50       | 0.13          |
> | Harvard-FairVLMed| 0.54           | 0.55           | 0.16         | 0.24    | 0.25  | **0.08**   | 0.25          |
>
>
> ***
>
> >**Q5**: Update and iteration of the evaluation benchmark.
>
> **A5**: With the rapid development in the field of LLMs, we will maintain CARES and update the results of the latest models on CARES as quickly as possible and expand the dataset and medical image modalities.
>
> ***
>
> >**Q6**: Limited clinical application guidance from evaluation results.
>
> **A6**: Our comprehensive evaluation of the trustworthiness of Med-LVLMs is not only aimed at promoting the development of Med-LVLMs but also at providing a reliable benchmark for medical experts, enabling them to select the appropriate model based on their needs. Inspired by your comments, we will introduce more detailed evaluation metrics, focusing on aspects that are of greater concern to clinical experts, thereby enabling a more fine-grained assessment of model trustworthiness.
>
> ***
>
> >**Q7**: Ignore the human-machine collaboration perspective.
>
> **A7**: Currently, CARES primarily focuses on assessing model performance. Evaluating model performance independently is the first step in measuring its potential capabilities. Evaluating human-machine collaboration is more complex and requires collaboration with clinical experts. In the future, we will assess the effectiveness of human-machine collaboration from various perspectives, such as designing experiments that simulate real medical scenarios, evaluating the model's performance in assisting doctors with decision-making, or introducing metrics to measure the model's impact on improving doctors' work efficiency.
>
> ***
>
> >**Q8**: Lack of consideration for long-term impacts.
>
> **A8**: In the future, we will incorporate long-term impact into our evaluation, such as by including the perspectives of more stakeholders, designing a more comprehensive evaluation framework, or conducting long-term follow-up studies to observe the profound effects of Med-LVLMs on the healthcare system.
>
> ***
>
> **Reference**:
>
> [1] DecodingTrust: A Comprehensive Assessment of Trustworthiness in GPT Models.

---

> > ### Author Response · Authors · 2024-08-22
> > **We would like to hear back from reviewer 8Xi4**
> >
> > Dear Reviewer 8Xi4,
> >
> > We wish to once again express our heartfelt appreciation for the time you took to review our paper.
> >
> > Given the short duration of the author-reviewer discussion period, we would greatly appreciate your feedback on whether your main concerns have been adequately addressed. We are always happy to provide further explanations or empirical results whenever necessary.
> >
> > Thank you very much!

---

### Official Review · Reviewer_UJHN · 2024-07-12
**CARES: A Comprehensive Benchmark of Trustworthiness in Medical Vision Language Models**

**Rating:** 7
**Confidence:** 4
**Correctness:** Yes
**Clarity:** Yes

**Review:**

Strong points of the paper:
1. CARES provides a multi-dimensional evaluation framework that is more holistic than previous works focusing on individual aspects of trustworthiness.
2. The use of 41K question-answer pairs across multiple modalities and anatomical regions ensures a broad and diverse evaluation.
3. The public release of the benchmark and code enhances reproducibility and allows other researchers to contribute to and validate the findings.

Weak points of the paper:
1. The paper highlights that all models evaluated show poor performance in multiple trustworthiness dimensions, which might indicate an issue with either the models or the robustness of the CARES benchmark.
2. The paper does not detail how the dataset avoids biases in its composition, which could affect the fairness evaluation.
3. The evaluation metrics, particularly for open-ended questions and uncertainty estimation, are complex and may not be easily reproducible or interpretable without significant expertise.

**Strengths:**

1. CARES provides a multi-dimensional evaluation framework that is more holistic than previous works focusing on individual aspects of trustworthiness.
2. The use of 41K question-answer pairs across multiple modalities and anatomical regions ensures a broad and diverse evaluation.
3. The public release of the benchmark and code enhances reproducibility and allows other researchers to contribute to and validate the findings.

**Additional Feedback:**

None

**Documentation:**

Yes

**Limitations:**

Yes

**Opportunities For Improvement:**

1. Case studies can be useful to provide some practical insights into trustworthiness.
2. It is helpful if the authors provide suggestions to improve trustworthiness of Med-LVLMs based on the observed deficiencies.

**Relation To Prior Work:**

Yes

**Summary And Contributions:**

The paper introduces a comprehensive benchmark called CARES to assess the trustworthiness of Med-LVLMs across five dimensions: trustfulness, fairness, safety, privacy, and robustness. CARES includes a dataset of 41K question-answer pairs across 16 medical image modalities and 27 anatomical regions. The study uncovers significant issues with Med-LVLMs, such as factual inaccuracies, lack of fairness across demographic groups, vulnerability to attacks, and privacy concerns.

---

> ### Author Rebuttal · Authors · 2024-08-17
>
> Thank you for your valuable feedback to help us improve our paper. We detail our response below and please kindly let us know if our response addresses your concerns.
>
> >**Q1**: The paper highlights that all models evaluated show poor performance in multiple trustworthiness dimensions, which might indicate an issue with either the models or the robustness of the CARES benchmark.
>
> **A1**: The evaluation methods for the five dimensions of trustworthiness follow common practice in LLMs [1], and we do not introduce any new metrics. Therefore, our evaluation results are reliable, and the existing trustworthiness issues in current Med-LVLMs are indeed significant, as also reflected in some recent studies [2, 3].
>
> ***
>
> >**Q2**: The paper does not detail how the dataset avoids biases in its composition, which could affect the fairness evaluation.
>
> **A2**: Since most medical image datasets, particularly multimodal datasets, do not include demographic information, it is challenging to account for the impact of demographic factors in our evaluations. Nevertheless, we incorporate four datasets containing demographic information (age, gender, race) into our evaluation benchmark, specifically MIMIC-CXR, Harvard-FairVLMed, HAM10000, OL3I. For these datasets, we have computed the demographic data distributions, which are presented in Figure 6 of the Appendix B (Page 3).
>
> ***
>
> >**Q3**: The evaluation metrics, particularly for open-ended questions and uncertainty estimation, are complex and may not be easily reproducible or interpretable without significant expertise.
>
> **A3**:
> - First, regarding the evaluation of open-ended questions, since open-ended VQA lacks precise evaluation metrics and common language quality evaluation metrics (e.g., BLEU, ROUGE) do not fully capture medical factuality, therefore, we adopt an automated metric [4] based on GPT-4 to score the ground-truth answers and model responses. To make the evaluation more reproducible, during the rebuttal period, we follow the POPE [5] by inputting medical reports into GPT-4 to convert them into closed-ended VQA, with answers limited to "yes" or "no." In summary, as shown in Table R1 in the attached PDF, all evaluated models still exhibit significant factuality hallucination.
>
> - For uncertainty estimation, following the approach in [6], we employ a prompt-based evaluation method, asking confidence-related questions to determine the model's uncertainty. Following your suggestion, during the rebuttal period, we adopt a probability-based approach for uncertainty estimation, using the model's produced conditional probabilities to determine its confidence levels. We then use ECE, AUROC, and MSE [7] as evaluation metrics. As shown in Table R2 in the attached PDF, the results align with the conclusions from the prompt-based testing approach, indicating that the current Med-LVLMs exhibit poorly calibrated uncertainty.
>
> ***
>
> >**Q4**: Case studies can be useful to provide some practical insights into trustworthiness.
>
> **A4**: We included some case studies in the attached PDF to help readers gain a more intuitive understanding of the trustworthiness of Med-LVLMs. And we will include more case studies in the Appendix of the next version to help readers better understand trustworthiness.
>
> ***
>
> >**Q5**: It is helpful if the authors provide suggestions to improve trustworthiness of Med-LVLMs based on the observed deficiencies.
>
> **A5**: To enhance trustworthiness across various dimensions, we can adopt different strategies. For example, to improve factuality, we can retrieve the medical descriptions most relevant to the input image as auxiliary prompts, or we can fine-tune the model using high-quality data for downstream tasks. For the improvement in safety, we can fine-tune the model using the data specifically designed for safety to align the behavior of the model. To improve fairness, we need to check the demographic information distribution of the training data and then fine-tune the model in the consideration of data distribution to avoid the group bias. The evaluation of trustworthiness in Med-LVLMs reveals that this is a highly promising area with significant room for improvement. We will include a more extensive discussion on methods to mitigate the trustworthiness issues of Med-LVLMs in the discussion section.
>
> ***
>
> **Reference**:
>
> [1] DecodingTrust: A Comprehensive Assessment of Trustworthiness in GPT Models.
>
> [2] Worse than Random? An Embarrassingly Simple Probing Evaluation of Large Multimodal Models in Medical VQA.
>
> [3] MedThink: Inducing Medical Large-scale Visual Language Models to Hallucinate Less by Thinking More.
>
> [4] LLaVA-Med: Training a Large Language-and-Vision Assistant for Biomedicine in One Day.
>
> [5] Evaluating Object Hallucination in Large Vision-Language Models.
>
> [6] Just Ask for Calibration: Strategies for Eliciting Calibrated Confidence Scores from Language Models Fine-Tuned with Human Feedback.
>
> [7] Semantic Uncertainty: Linguistic Invariances for Uncertainty Estimation in Natural Language Generation.

---

### Decision · Program_Chairs · 2024-09-26

**Decision:**

Accept (Poster)

**Comment:**

This paper comprehensively evaluates the Trustworthiness of Med-LVLMs across the medical domain. It evaluates the trustworthiness of Med-LVLMs across five dimensions, including trustfulness, fairness, safety, privacy, and robustness. All reviewers agree to accept the paper due to its comprehensiveness and reproducibility. I vote for an acceptance.